# The Effects of COVID-19 Lockdown on Social Connectedness and Psychological Distress in U.S. Adults with Chronic Diseases

**DOI:** 10.3390/ijerph20136218

**Published:** 2023-06-24

**Authors:** Ben King, Omolola E. Adepoju, LeChauncy Woodard, Abiodun O. Oluyomi, Xiaotao Zhang, Christopher I. Amos, Hoda Badr

**Affiliations:** 1Department of Health Systems and Population Health Sciences, College of Medicine, University of Houston, Houston, TX 77204, USA; oadepoju@central.uh.edu (O.E.A.); lwoodard@central.uh.edu (L.W.); 2Humana Integrated Health Systems Sciences Institute, University of Houston, Houston, TX 77204, USA; 3Section of Epidemiology and Population Sciences, Department of Medicine, Baylor College of Medicine, Houston, TX 77030, USA; abiodun.oluyomi@bcm.edu (A.O.O.); chris.amos@bcm.edu (C.I.A.); hoda.badr@bcm.edu (H.B.); 4Institute for Translational Epidemiology & Division of Liver Diseases, Department of Medicine, Icahn School of Medicine at Mount Sinai, New York, NY 10029, USA; xiaotao.zhang@mountsinai.org

**Keywords:** COVID-19, SARS-CoV-2, chronic disease, chronic illness, social connectedness, psychosocial, psychological distress

## Abstract

Lockdown measures enacted in 2020 to control the spread of COVID-19 led to increases in the prevalence of mental health problems. Due to their high-risk status, individuals with chronic diseases may be at increased risk and disproportionately adversely affected by the COVID-19 pandemic. The investigators examined associations between having a high-risk chronic condition, social connectedness, and general distress and COVID-19-specific distress among U.S. adults during the COVID-19 lockdown. Baseline measures of a longitudinal survey collected at the beginning of the pandemic (April to June 2020) were analyzed to identify factors associated with loss of social connectedness from pre- to post-lockdown. The associations between social connectedness and both general and COVID-19-specific psychological distress were adjusted for certain high-risk chronic illnesses and interaction effects. The sample available for analysis included 1354 subjects (262 high-risk chronic diseases and 1092 without chronic illness). Those reporting the loss of social connectedness were younger (median = 39 vs. 42) and more likely to be unemployed because of the pandemic (19.4% vs. 11.0%). Adjustment for interaction demonstrated a stronger negative association between social connectedness change and the psychosocial impact of COVID-19 for those with high-risk illness(es) (change in connectedness*chronic illness OR = 0.88, 95%CI: 0.79–0.98, *p* = 0.020). These findings inform our understanding of the distribution and intersection of responses to public health lockdown orders in the U.S. and build further evidence of the importance of social connectedness on psychological distress.

## 1. Introduction

Coronavirus disease 2019 (COVID-19) caused by the SARS-COV-2 (severe acute respiratory syndrome coronavirus 2) virus was an unprecedented public health crisis. Governments across the globe initially responded by instituting social distancing guidelines and strict lockdowns involving mandatory stay-at-home orders and the temporary closure of non-essential businesses and schools [1,2]. While such measures were critical for mitigating virus transmission, they disrupted all aspects of social life with far-reaching impacts on the economy, social relationships, and physical and mental health with the potential to endure long after they were repealed [3,4,5,6,7,8]. Understanding the effect of these measures, particularly on vulnerable populations, offers insight into the broader damage caused by the pandemic. It also provides critical information for public health leaders and policy makers looking ahead to the next public health crisis.

Humans are social beings by nature, and the loss of positive social connections and relationships can have profound effects on mental health outcomes [3]. When lockdown measures were enacted to control the spread of COVID-19, significant increases in the prevalence of mental health problems were documented worldwide, ranging from feelings of low mood and worry to clinically significant levels of depression, anxiety, and even suicidality [4,5,6,7]. These findings led some to postulate retrospectively that decrements in social connectedness due to COVID-19 lockdown measures may be driving these effects [3,8]. Social connectedness refers to the sense of belonging and subjective psychological bond that people feel in relation to individuals and groups of others [9]. Research has shown that having close and more diverse social connections is associated with a lower risk of depression, greater emotional well-being, and better physical health [10,11,12]. In contrast, having fewer and less diverse social connections is associated with poorer mental [13] and physical health [14,15,16] and has even been linked to early mortality [17].

Even though the COVID-19 pandemic and associated lockdown affected the entire population, it did not affect all individuals equally. For example, the odds of experiencing depression and anxiety symptoms during the COVID-19 lockdown were 2.5 times higher for low-income individuals compared with high-income individuals [18]. Individuals with pre-existing mental health problems were also more likely to report increased depression, anxiety, and post-traumatic stress disorder symptoms during lockdown than the general population [19,20,21]. Thus, certain population subgroups appear more vulnerable to the adverse mental health effects of the loss of social connectedness caused by virus mitigation measures. Because the mental health consequences of the COVID-19 pandemic are projected to be responsible for its most pervasive and enduring health footprint [22], research that further elucidates the impact of social connectedness patterns on lasting mental health outcomes is critical. Addressing these issues may help to inform a more effective, targeted approach to the ongoing and any future public mental health responses by stimulating discourse on the types of supports that may be needed for individuals from vulnerable subgroups during this and future pandemics.

During the early stages of the pandemic, there was limited investigation into the incipient and unfolding mental health impacts of COVID-19 lockdown policies, particularly on individuals with chronic diseases [23]; however, research on the effects of quarantine during the Middle East Respiratory Syndrome (MERS) outbreak found that these individuals had elevated clinical anxiety and depression levels [24] and increased odds for clinically significant anxiety 4–6 months post quarantine [25]. Since that first year of the pandemic response, multiple studies have shown that individuals with chronic diseases may be an especially vulnerable population for mental health sequelae [26,27,28]. However, more research on the effects of the COVID-19 pandemic on their mental health is needed [26].

For several reasons, individuals with chronic diseases may be at increased risk for decrements in social connectedness, and their mental health may be disproportionately adversely affected by the COVID-19 pandemic. First, given the ongoing social and emotional challenges of chronic diseases and their treatment [29,30], these individuals are already at increased risk of comorbid depression relative to the general population [29,30,31,32]. Second, there are some “high-risk” chronic conditions (e.g., diabetes, heart disease, chronic lung diseases) where individuals are more susceptible to severe COVID-19 and death [33,34], and this may contribute to elevated anxiety levels. Third, perceptions of increased health risks from COVID-19 may motivate adherence to mitigation strategies, thus increasing the likelihood of social isolation. Fourth, pandemic-related financial hardship [35,36,37] could affect their ability to pay for chronic illness-related medical and psychosocial care. Finally, in the early days of the pandemic, hospitals prioritized COVID-19 patients, leading to treatment delays and interruptions for individuals with chronic conditions [38], which could have exacerbated their distress. However, the effects of lockdown on COVID-19-specific distress and factors that may contribute to elevated distress levels in this population have yet to be thoroughly examined. Moreover, many studies examining the mental health effects of pandemics and other public health crises have focused almost exclusively on general (psychiatric) measures of distress [3]. Such measures may not be sensitive enough to capture the direct negative psychosocial impacts of these traumatic events, resulting in possible underestimation of their true impact on population-level mental health.

### Research Hypotheses

Given the above, this case–control study examined associations between having a high-risk chronic condition, social connectedness, and general and COVID-19-specific distress among U.S. adults during the COVID-19 lockdown. Specifically, we examined (1) the characteristics associated with loss of social connectedness, (2) the strength of associations between mental health and changes in social connectedness, and (3) whether the pandemic differentially affected those associations between mental health and social connectedness in individuals with high-risk chronic diseases. These findings will hopefully provide useful information for intervention developers on whether greater social connectedness may help to buffer these individuals from the adverse effects on their distress and mental health from this and any future public health lockdowns.

## 2. Materials and Methods

This study reports on baseline data obtained from the initial survey of a longitudinal study on the psychosocial and health-behavioral impacts of the COVID-19 pandemic [39]. Eligible individuals were ≥age 18, who resided in the United States, were fluent in English or Spanish, and reported living under stay-at-home orders on the questionnaire. Electronic surveys were administered on the Qualtrics platform (Provo, UT, USA) in English and Spanish. They were distributed via paid and unpaid social media advertisements and an online survey crowdsourcing platform (Soapbox Sample) between 13 April and 8 June 2020. This recruitment window corresponded to the initial stay-at-home order period adopted in most of the United States. Recruitment advertisements and social media posts contained a web hyperlink that directed participants to the survey landing page, which contained a brief cover letter describing the purpose of the research, eligibility criteria, and a plain language statement. If, after reading the cover letter, individuals were interested in participating, they checked a box to confirm understanding and consent to participate. For quality-control purposes, we employed attention checks (e.g., red-herring questions), tactics to help prevent machine responses (e.g., Captcha), I.P. control to ensure individuals could not take the survey more than once, and data quality checks (e.g., answer consistency and speed checks).

### 2.1. Survey Measures

#### 2.1.1. Sociodemographic and Health-Related Variables

Sociodemographic variables included reports of respondents’ age, gender, race/ethnicity, education, marital status, household income, number of household residents, whether they lived with someone over 65 (yes/no) or under 18 (yes/no), employment status, and residence zip code and nearest cross streets. Based on the latter, we used the 2010 Rural-Urban Commuting Area (RUCA) codes to define urban and rural areas of residence [40,41]. RUCA codes classify census tracts based on population density, urbanization, and daily commuting flow measures.

Health-related variables included chronic illness status and adherence to COVID-19 social distancing and lockdown measures. Concerning the former, individuals reported whether they had a chronic or serious health condition that required medication or management at home (yes/no) and, if so, were asked to specify the condition. To assess adherence with stay-at-home orders, we first asked, “Is the area where you live currently under a ‘Stay-at-Home,’ ‘Safer-at-Home,’ or ‘Shelter-at-Home’ order? (yes/no)”. The sample was restricted to respondents who replied “yes” to this question. We then asked, “To what extent do you currently follow the stay-at-home order?” With regard to social distancing, we asked, “What amount of social distancing do you currently practice?” With regard to other public health behaviors, we asked, “Do you currently practice protective measures?” Response options to all three behavior questions were on an 11-point Likert-type scale from 0 to 10, with higher scores indicating greater endorsement or frequency.

#### 2.1.2. Social Connectedness

The survey asked, “Before/Since the COVID-19 pandemic, how many individuals did you feel close to who live outside your household?” Response options were on an 11-point Likert-type scale from 0 (none) to 11 (10 or more). Change in social connectedness (i.e., delta) was computed by subtracting the score prior to the pandemic from the score since the pandemic (range of −10 to 10). This range was then dichotomized to indicate a loss of social connectedness related to the pandemic (1 = negative scores indicating loss of social connections, 0 = non-negative scores indicating no change or an increase in social connections).

#### 2.1.3. Psychological Distress Outcome Measures

General distress was assessed using the 4-item short-form Patient-Reported Outcome Measure Information System (PROMIS) depression and anxiety measures [42,43]. The depression short-form captures a respondent’s negative mood and views of the self, and the anxiety short-form assesses fear, worry, and hyperarousal over the past 7 days. Responses for both measures range from 1 (never) to 5 (always) and are summed to form a raw score that can then be scaled into a T-score (nationally standardized) with a mean of 50 and a standard deviation of 10. T-scores > 60 indicate the need for further psychological evaluation [44,45].

COVID-19-specific distress was assessed using the 4-item short form PROMIS Psychosocial Impact of Illness—Negative (PII-N) scale [46,47]. Instead of asking about illness, we prompted individuals to think about the direct impacts of COVID-19 “before the COVID-19 pandemic” and “since the COVID-19 pandemic” on a scale from 1 (not at all) to 5 (very much). Based on the PROMIS scoring instructions, a raw score is calculated by summing the “since the COVID-19 pandemic” responses and then scaled into a T-score (standardized) with a mean of 50 and a standard deviation of 10 [44,45].

### 2.2. Statistical Analysis

Descriptive statistics were calculated to characterize the sample, including the mean, standard deviation (S.D.), median, range (for continuous variables), and relative frequency (for categorical variables). Subsequent analyses focused on bivariate and multivariate associations between changes in social connectedness due to the pandemic and either general or COVID-19-specific psychological distress.

This process entailed two steps. In Step 1, we identified factors associated with losses in social connectedness for the overall sample using bivariate tests of statistical significance (chi-squared tests for categorical and Wilcoxon ranksum tests for continuous independent variables) and effect size (odds ratios and 95% confidence intervals). In Step 2, multivariate logistic regression analysis was conducted to estimate the association between each of the psychological distress outcome variables (general and COVID-19 specific) and chronic illness status, change in social connectedness, and the interaction between chronic illness status and social connectedness change. All analysis was conducted using STATA v16.1 (College Station, TX, USA).

## 3. Results

### 3.1. Sample Characteristics

Of the 2435 adults who consented to participate, 213 were excluded because they did not pass our survey quality control (i.e., re-captcha, red-herring questions, I.P. control) and data quality checks (e.g., answer consistency and speed checks). Of the 2222 survey respondents who provided usable data, 1825 responded to the stay-at-home question and 1671 (91.5%) indicated that they were currently under Stay-at-Home orders at the time of the survey. Of those, 262 indicated that they had one of the high-risk chronic diseases of interest (i.e., chronic lung disease, diabetes, heart disease) and 1092 reported that they did not have a chronic illness. Thus, the study sample comprised 1354 respondents.

Table 1 presents summary measures of sociodemographic characteristics, connectedness, health and behaviors, and psychological distress. The median age was 40.5 years (32.7% over 50 years), and the majority were female (67.6%), White (61.7%), married (55.2%), and college-educated (74.63%). Respondents also reported high levels of following stay-at-home orders, social distancing, and other protective measures (median = 9 on a 0–10 scale for all 3).

Overall, 40.1% of the sample reported a loss of social connectedness, 52.9% reported no change, and 7.9% reported increases in social connectedness (Figure 1). Median scores on the PROMIS measures (median = 55.7, 59.5, and 55; lower bound limit of IQR = 49, 53.7, and 50.3, respectively) were above nationally standardized means, demonstrating disproportionately high levels of psychological distress. Across the full cohort, the case threshold (T score > 60) was met for depression in 31.2%, anxiety in 47.8%, and the psychosocial impact of COVID-19 in 23.6% of cases.

### 3.2. Associations with Loss of Social Connectedness

Table 2 shows bivariate relationships between survey measures and reported loss of social connectedness. Overall, those reporting the loss of social connectedness were younger (median = 39 vs. 42), less likely to be married (50.8% vs. 58.1%) or college educated (71.5% vs. 76.8%), and more likely to be unemployed as a result of the pandemic (19.4% vs. 11.0%). They also reported slightly lower levels of social distancing (OR = 0.93; 95% CI: 0.87–1.00).

### 3.3. Associations between Changes in Loss of Social Connectedness and Psychological Distress

Consistent, strong associations between a loss of social connectedness and depression and anxiety were found (Table 2). Individuals who reported the loss of social connectedness had two-fold greater odds of depression (OR = 2.18; 95% CI: 1.69–2.80, *p* < 0.001) and anxiety (OR = 2.07; 95% CI: 1.63–2.62, *p* < 0.001); and an 81% greater odds of reaching the case threshold for COVID-19 specific distress (95% CI: 1.38–2.37; *p* < 0.001).

### 3.4. Multivariate Logistic Regression Results

#### 3.4.1. General Psychological Distress

As Table 3 shows, lower social connectedness was associated with greater odds of meeting the case threshold for depression (OR = 0.90; 95% CI: 0.85–0.94; *p* < 0.001) and anxiety (OR = 0.91; 95% CI: 0.87–0.96; *p* < 0.001).

#### 3.4.2. COVID-19 Specific Distress

Adjustment for chronic illness status did not alter the significance of the association between close social connectedness and PII-N (unadjusted-OR = 0.95; 95% CI: 0.90–0.99 vs. aOR = 0.95; 95% CI: 0.91–0.99). However, the lone instance of interaction between chronic illness and social connectedness demonstrated a stronger negative (i.e., inverse) association between social connectedness change and psychosocial impact of COVID-19 for those with a chronic condition (interaction between close contact loss*chronic illness OR = 0.88, 95%CI: 0.79–0.98, *p* = 0.020; Table 3, Figure 2). Stratification of the relationship by chronic illness further illustrates the stronger association between COVID-19-specific distress and the loss of social connectedness for those with chronic illnesses (OR = 3.1; 95% CI: 1.7–5.9 vs. OR = 1.6; 95% CI: 1.7–2.1; Mantel–Haenszel test for homogeneity: *p* = 0.49).

## 4. Discussion

This baseline survey establishes several important findings, including identifying certain characteristics associated with losing social connectedness in the earliest days of the pandemic. Around one-third of the participants reported losing close contacts in their social network over a matter of weeks or months from before the outbreak to the time they completed the survey, demonstrating the serious scale of the impact, while also suggesting that the social impacts of lockdown were not distributed evenly across the U.S. It is this particular social construct (decrease in reported social connectedness) which became the focus for this analysis. In addition, the levels of psychosocial distress one standard deviation above a population-standardized mean (T-score > 60; 31.2% depression; 47.8% anxiety, 23.6% psychosocial impact of COVID-19) in this sample indicates a high level of need for psychosocial support during the time this survey was collected. Finally, survey respondents demonstrated a strong, consistent association between poorer general and COVID-19-specific psychological distress and reduced social connectedness.

The lone statistical interaction effect showed a larger tradeoff between loss of close social connectedness and a measure of the emotional and psychological toll (PII-N) specifically related to the pandemic (Table 3; Figure 2). While prior research has shown that people living at higher risk from their chronic illness were more likely to experience mental health problems [23,24,26,28], this study shows that the loss of contacts they felt close to was associated with even greater feelings of worthlessness, disconnection, worry, or lacking meaning in life (PII-N items) related to COVID-19 than the respondents without chronic illness. Notably, this finding was limited to COVID-19-specific psychosocial distress, while those with chronic illnesses tended to mirror the healthier respondents regarding their general psychological distress response. This also makes intuitive sense that respondents with high-risk illnesses, knowing little about the outbreak in these early weeks except that they were at higher risk of death, might have a stronger connection between decreases in close contacts at the time and increased psychosocial impact, which they attributed to the pandemic.

These findings are notable for providing hindsight into the experiences felt in the early weeks and months of the pandemic. Perhaps more importantly, they also provide useful information for programs responding to future public health crises or addressing the more generalized, ongoing problems of social connectedness and psychological distress. These findings also emphasize the importance of developing treatment plans to improve the community’s response to future public health crises. While the SARS-CoV-2 pandemic was a generational public health disaster, there is also reason to believe that the globe will face more such events in coming years [48]. Such plans should incorporate screening and access to treatment in order to mitigate general and context-specific psychosocial repercussions, with special attention given to those individuals afflicted by chronic illnesses that are associated with increased health risks. Previous research suggests that a public health crisis, especially when many individuals are experiencing social isolation, can have secondary effects, such as negative mental health outcomes that could be alleviated through focused interventions [3,49,50,51]. For example, disseminating accurate and up-to-date information about the virus is vital so that individuals feel more in control of the situation and know how to protect themselves [49]. Of note, those who lost social connectedness during lockdown reported similar levels of adherence to stay-at-home orders and to use of other protective behaviors but slightly lower social distancing practices (Table 2). Of course, the self-reported difference in behavioral adherence may reflect a real difference in behaviors or a differential in awareness of their own practice or perceptions of the optimal degree of adherence.

While no effect was identified between loss of contacts and gender or race in this survey, the study indicated that those with a loss of connectedness were less likely to be married, less likely to have a college education, and less overall and COVID-19 specific unemployment. This result suggests there may be other underlying social and economic supports which predict changes to individuals’ social connectedness during the pandemic and perhaps influence the exacerbation or mitigation of general and disease-specific psychological distress responses.

Based on these findings, the specific impact of distress from the pandemic, lockdowns, and resulting loss of social connectedness on the needs of those with chronic diseases and what interventions are needed most should be further investigated [52]. Emphasis should be placed on monitoring mental health during periods of isolation such as was encountered when the COVD-19 lockdown disturbed social networks. Primary healthcare professionals should be educated and supported in screening for these symptoms and providing or referring to services to help their patients manage their distress and general mental health during a lockdown [53]. Tailored instruction and strategies to screen and monitor mental health through online and telehealth modalities could be developed and implemented for providers and other healthcare professionals [54,55]. Alternative strategies to reach those individuals without reliable internet access should also be explored to avoid the further exacerbation of current health disparities [52].

The long-term effects of COVID-19 are still unknown, but these findings and previous evidence [3,4,5,6,7,8] indicate how vital it is to care not only for the physical health of individuals during a pandemic but also for their mental health. This need starts from the beginning of the social isolation period. Based on these findings, responses to future pandemics should consider including interventions to address the general and contextualized psychosocial impacts, with particular focus on identifying and screening those individuals living with high-risk chronic illnesses.

### Limitations

As an analysis of a baseline survey, temporality is difficult to establish with this sample and the reported findings represent correlations, not causal relationships. While some items specifically focus on changes due to the onset of the pandemic or community lockdowns, others, such as the general distress measures of depression and anxiety, do not necessarily account for the timing of symptom onset or exacerbation from the pandemic. The self-reported chronic illnesses may be presumed to identify diagnoses that predate the pandemic; however, this is not specifically documented. Both the change in social connectedness assessed pre- and post-lockdown, albeit via recall, and the PII-N measure were worded to assess the impact of COVID-19 specifically.

In addition, the entire survey is based on self-reported answers that pose a threat of response bias. For example, those respondents reporting chronic diseases may also present a threat of biased recall if they devoted greater consideration to their social interactions, exposures, or psychological distress effects than other participants.

While the study reports findings collected from respondents across the U.S., the participants do not necessarily reflect a representative sample. Demographics, living conditions, and family sizes all differ to varying degrees from national averages. This naturally impacts the generalizability of the findings, which should be interpreted in the context of this sample. Bivariate and multivariate testing also reflect an available case analysis, with listwise deletion of cases with missing values. For this reason, the sample sizes for individual tests are reported in the tables to document the sample size included in each instance appropriately.

The chronic illness group is smaller than the healthier comparison group and not a monolith itself. Diabetes and heart and lung diseases are very different conditions with different management needs. They are collapsed in this analysis to reflect the elevated risk of severe illness posed by these conditions, but larger samples of those living with these chronic diseases should be collected to understand better the heterogeneity of any effects reported here. Finally, the analysis also involves multiple tests for statistical significance with a standard type I error rate of 5%, indicating some degree of risk for false positive results without adjustment. However, the principal findings of the dynamic relationships between social connectedness, distress, and chronic illness within this study are shown to be consistent through multiple testing approaches (Table 2 and Table 3).

## 5. Conclusions

The loss of connectedness in one’s social network was heavily associated with poorer psychological distress and mental health during the early months of the 2020 pandemic. The association between the loss of close contacts since lockdowns and COVID-19-specific distress (PII-N) was stronger for those respondents living with and managing chronic illnesses that put them at higher risk for severe COVID-19. A loss of contacts was also associated with living situation and unemployment during the pandemic, among other things. These findings inform our understanding of the distribution and intersection of responses to public health lockdown orders in the U.S. They also build further evidence of the importance of social connectedness on psychological distress more generally. In the event of ongoing or new public health crises, psychosocial interventions targeting individuals identified as high-risk for disease and death should be developed and rapidly implemented.

## Figures and Tables

**Figure 1 ijerph-20-06218-f001:**
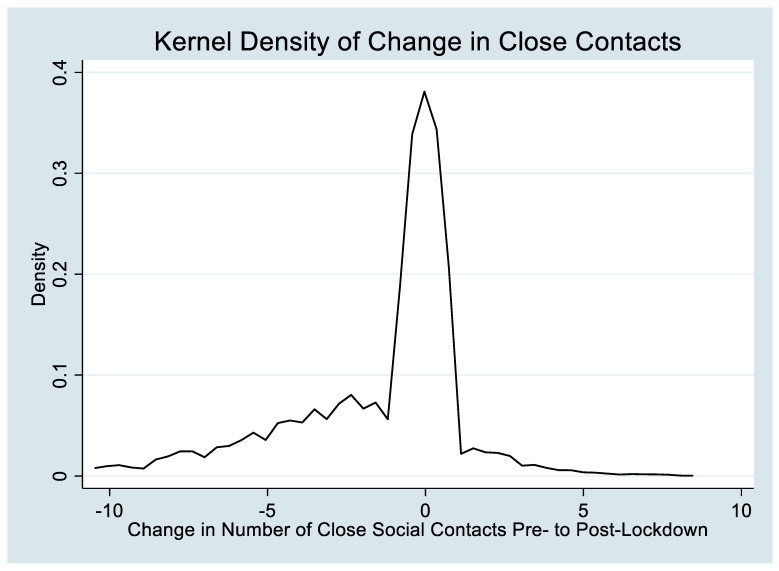
Density plot of change in close contacts from pre- to post-lockdown.

**Figure 2 ijerph-20-06218-f002:**
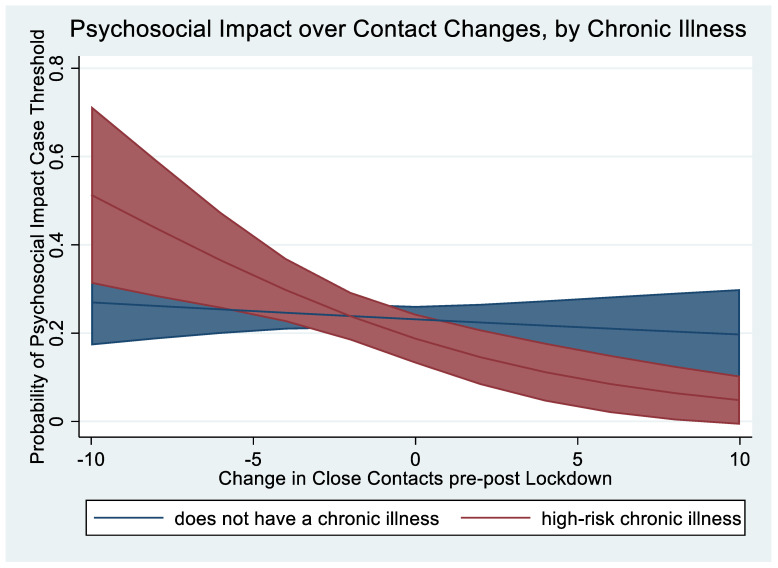
Margin plot of the Negative Psychosocial Impact related to COVID-19 (PII) over the change in close contacts reported {−10,10}, by chronic illness status.

**Table 1 ijerph-20-06218-t001:** Sample characteristics and survey responses.

		n	n	%
			/Median	/IQR
Demographics and Living Situation				
	Age {18–93}	1334	40.5	(30–54)
	% Elderly (>=50)	1334	436	32.68%
	% Male	1350	437	32.37%
	% Married	1351	746	55.22%
	% Non-White	1341	514	38.33%
	% College Educated	1352	1009	74.63%
	Household Income {1–5}	1308	$75,000–$99,999	($25k–$75k–$150k or more)
	Has your work status changed as a result of COVID? (Y/N)	1353	579	42.79%
	Reduced Hours or Income Due to COVID-19? (Y/N)	1348	98	7.27%
	Unemployed Due to COVID-19? (Y/N)	1028	141	13.72%
	Working full or part-time (Y/N)	1349	957	70.94%
	Unemployed, NOT retired (Y/N)	1349	263	19.50%
	How many people live with you now (including yourself)?	1346	3	(2–4)
	Family over 65 (Y/N)	1347	231	17.15%
	Family under 18 (Y/N)	1349	501	37.14%
	Living in a Rural Zip (Y/N)	1327	87	6.56%
Social Connectedness				
	Change in Social Connectedness {−10,10}	1216	0	(−3–0)
	Loss of Social Connectedness (Y/N)	1216	488	40.13%
Health				
	Chronic illness status (Y/N)	1354	262	19.35%
	To what extent do you currently follow the stay-at-home order?	1351	9	(8–10)
	What amount of social distancing do you currently practice?	1337	9	(8–10)
	Do you currently practice other protective measures?	1338	9	(7–10)
Mental Health (PROMIS and PSS)				
	PROMIS Depression (T Score)	1259	55.7	(49–62.2)
	PROMIS Depression Case (Y/N)	1259	393	31.22%
	PROMIS Anxiety (T Score)	1253	59.5	(53.7–65.3)
	PROMIS Anxiety Case (Y/N)	1253	599	47.81%
	PROMIS Psychosocial Impact of Illness (T Score)	1241	55	(50.3–59.3)
	PROMIS Psychosocial Impact of Illness Case (Y/N)	1241	293	23.61%

**Table 2 ijerph-20-06218-t002:** Sample characteristics and survey responses by loss of social connectedness pre–post lockdown.

			No Loss (0 or +)		Loss of Connectedness (−)			
			n = 728		n = 488			
		n	n	%	n	%	OR	95% CI
			Median	IQR	Median	IQR		
	Age {18–93}	1201	42	(32–58)	39	(29–51)	0.984 *	0.977–0.991
	% Elderly (>=50)	1201	268	37.43%	139	28.66%	0.672 *	0.523–0.862
	% Male	1214	243	33.43%	148	30.39%	0.870	0.679–1.114
	% Married	1214	422	58.13%	248	50.82%	0.744 *	0.591–0.938
	% Non-White	1205	259	35.97%	192	39.59%	1.166	0.920–1.479
	% College Educated	1215	558	76.75%	349	71.52%	0.760 *	0.585–0.988
	Household Income {1–5}	1175	3	(2–5)	3	(2–4)	0.924	0.851–1.004
	Has your work status changed as a result of COVID? (Y/N)	1215	437	60.11%	245	50.20%	0.669 *	0.530–0.844
	Reduced Hours or Income Due to COVID-19 (Y/N)	1210	50	6.91%	40	8.23%	1.209	0.784–1.864
	Unemployed Due to COVID (Y/N)	913	60	10.97%	71	19.40%	1.954 *	1.342–2.844
	Working full or part-time (Y/N)	1211	519	71.59%	339	69.75%	0.915	0.711–1.178
	Unemployed, NOT retired (Y/N)	1211	117	16.14%	115	23.66%	1.611 *	1.20–2.151
	How many people live with you now (including yourself)?	1210	2.5	(2–4)	3	(2–4)	1.023	0.946–1.107
	Family over 65 (Y/N)	1211	139	19.12%	78	16.12%	0.813	0.599–1.103
	Family under 18 (Y/N)	1212	249	34.25%	188	38.76%	1.215	0.957–1.543
	Living in a Rural Zip (Y/N)	1201	41	5.72%	39	8.06%	1.445	0.917–2.278
Health and Behaviors								
	Chronic illness status (Y/N)	1216	146	20.05%	100	20.49%	1.027	0.773–1.366
	To what extent do you currently follow the stay-at-home order?	1351	9	(8–10)	9	(8–10)	0.960	0.897–1.027
	What amount of social distancing do you currently practice?	1337	9	(8–10)	9	(8–10)	0.933 *	0.869–1.000
	Do you currently practice other protective measures?	1338	9	(7–10)	9	(7–10)	0.990	0.935–1.048
Mental Health (PROMIS & PSS)								
	PROMIS Depression (T Score)	1208	53.9	(41–58.9)	58.9	(51.8–63.9)	1.055 *	1.041–1.069
	PROMIS Depression Case (Y/N)	1208	177	24.55%	202	41.48%	2.178 *	1.694–2.802
	PROMIS Anxiety (T Score)	1207	57.7	(51.2–63.4)	61.4	(57.7–67.3)	1.053 *	1.039–1.067
	PROMIS Anxiety Case (Y/N)	1207	292	40.50%	284	58.44%	2.066 *	1.629–2.620
	PROMIS Psychosocial Impact of Illness (T Score)	1205	54	(48.7–58.5)	56.8	(52.9–60.1)	1.063 *	1.045–1.082
	PROMIS Psychosocial Impact of Illness Case (Y/N)	1205	139	19.28%	146	30.17%	1.809 *	1.380–2.370

* *p* < 0.05, derived from Wilcoxon rank-sum or chi-squared tests (not odds ratios).

**Table 3 ijerph-20-06218-t003:** Models of psychosocial distress—social connectedness change, chronic illness status, and their interaction.

General and COVID-Specific Distress Case Thresholds				Adjusted for Chronic Illness			Interaction: Chronic Illness*Change in Social Connectedness		
			n	OR	Lower Limit	Upper Limit	OR	Lower Limit	Upper Limit
		Loss of Connectedness Pre-Post Lockdown (y/n)							
PROMIS Depression (case)		Change in Social Connectedness {d = −10,10}	1208	0.900 *	0.861	0.940	0.896 *	0.852	0.942
		Chronic Illness (y/n)		1.016	0.750	1.377	1.045	0.733	1.492
		Change in Social Connectedness*Chronic Illness					1.016	0.917	1.126
PROMIS Anxiety (case)		Change in Social Connectedness {d = −10,10}	1207	0.907 *	0.869	0.947	0.910 *	0.867	0.956
		Chronic Illness (y/n)		0.890	0.669	1.183	0.870	0.628	1.205
		Change in Social Connectedness*Chronic Illness					0.986	0.891	1.090
PROMIS Psychosocial Illness Impact (case)		Change in Social Connectedness {d = −10,10}	1205	0.949 *	0.905	0.995	0.980	0.927	1.035
		Chronic Illness (y/n)		0.987	0.709	1.374	0.767	0.511	1.150
		Change in Social Connectedness*Chronic Illness					0.877 *	0.785	0.979

* *p* < 0.05. Models are unadjusted for any variables beyond those listed (Change in Social Connectedness, Chronic Illness, and their Interaction).

## Data Availability

Data are available on request due to privacy restrictions. The data presented in this study are available on request from Hoda Badr (hoda.badr@bcm.edu).

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
