# Peer review of "The Effects of COVID-19 Lockdown on Social Connectedness and Psychological Distress in U.S. Adults with Chronic Diseases"

_ijerph, 2023, doi:10.3390/ijerph20136218_

Round 1

Reviewer 1 Report

I thank the authors for the interesting manuscript, but suggest some changes to improve it. In particular:

-I recommend making the research hypotheses more explicit with a separate section.

-Tables are too chaotic, values entered are often redundant.

-No need to represent significances with a different color, but with asterisk or other symbols.

-I also recommend reviewing the figures and the explanation of them.

-In light of the hypotheses identified, I recommend discussing the results more systematically.

-I recommend including practical applications of the study.

Best regards

Author Response

Thank you for the thoughtful feedback on our manuscript. We have done our best to incorporate each concern you mentioned. Our responses follow each recommendation.

I recommend making the research hypotheses more explicit with a separate section.

  • Done

Tables are too chaotic, values entered are often redundant.

  • The tables were reduced where redundancy was identified. P-values were removed, ORs & CIs retained.

No need to represent significances with a different color, but with asterisk or other symbols.

  • Asterisks indicating significance were applied to OR column.

I also recommend reviewing the figures and the explanation of them.

  • Reviewed and explanations were clarified

In light of the hypotheses identified, I recommend discussing the results more systematically.

  • The hypotheses were ordered to reflect the order in which the results were presented. Results were also reorganized slightly, but consistency with order of tables was maintained.

I recommend including practical applications of the study.

  • The current statement (3rd and 6th paragraph of Discussion) was strengthened regarding the applicability of these findings to current mental health interventions focused on the COVID-19 lockdown and also for future pandemics.

Reviewer 2 Report

Dear authors,

please see the attached file with my comments.

Author Response

Thank you for the thoughtful feedback on our manuscript. We have done our best to incorporate each concern you mentioned. Our responses follow each recommendation.

Abstract

line 22: “…1,354 surveys….” Do you mean subjects?

  • Corrected

  1. Introduction

lines 39-42: “While such measures…after they were repealed”. Authors, please provide a reference for this statement.

  • Moved up references 3-8 to support this sentence.

lines 67-70: “Because the mental health consequences… in addressing mental health inequities”. It is not clear what authors mean by this sentence, especially on how social connectedness is related with the long-term effects of COVID-19. Please, rephrase.

  • Rephrased

line 81: Delete “and”.

  • Rephrased

lines 98-100: “Moreover, many studies…. Measures of distress.” Please provide one re two indicative references, especially for “other public health crises”.

  • Added Brooks et al. [3] to support the statement

  1. Materials and Methods

lines 144-145: Which were the response options to the social distancing question “What amount of social distancing do you currently practice?”

  • Following the listing of the 3 behavioral questions: ‘follow stay-at-home order’, ‘social distancing practice’, and ‘protective measures practice’, we state that the response ranges for all three questions was {0-11}.

  1. Discussion

Lines 277-279: “While prior research has shown …. mental health problems….”. Authors, please provide a reference for this statement.

  • Added Louvardi et al. [23], Lee et al. [24], Saqib et al. [26], and Park & Park [28] to support statement

Line 303: Please replace “sex” with “gender”

  • Corrected

Lines 306-309: “this result suggests…. psychological distress responses”. Please provide evidence for this notion especially in relation to the role social support and resilience.

  • Thank you for this very thoughtful question. The statement is in reference to the evidence mentioned immediately prior: loss of connectedness was associated with marital status, education level, and employment status. It is intended as a statement that these resources may be associated with changes to connectedness and may also be associated with distress responses (i.e. potential confounders). This sentence was rephrased to hopefully make that intent clearer and also remove “resilience” from the equation.

Lines 322-323: “…and previous evidence…” Authors, please provide a reference for this

statement.

  • Added

Round 2

Reviewer 1 Report

The manuscript is definitely improved, my advice is always to improve the figures as quality, but the journal will point you out.